

# Prediction of HIV-1 protease resistance using genotypic, phenotypic, and molecular information with artificial neural networks

Huseyin Tunc[1], Berna Dogan[2], Büşra Nur Darendeli Kiraz[3,4], Murat Sari[5], Serdar Durdagi[6,7] and Seyfullah Kotil[3,8]

[1] Department of Biostatistics and Medical Informatics, School of Medicine, Bahcesehir University, Istanbul, Turkey
[2] Department of Medicinal Biochemistry, School of Medicine, Bahcesehir University, Istanbul, Turkey
[3] Department of Biophysics, School of Medicine, Bahcesehir University, Istanbul, Turkey
[4] Department of Bioengineering, Yildiz Technical University, Istanbul, Turkey
[5] Department of Mathematics Engineering, Faculty of Science and Letters, Istanbul Technical University, Istanbul, Turkey
[6] Computational Biology and Molecular Simulations Laboratory, Department of Biophysics, School of Medicine, Bahcesehir University, Istanbul, Turkey
[7] Department of Pharmaceutical Chemistry, School of Pharmacy, Bahcesehir University, Istanbul, Turkey
[8] Department of Molecular Biology and Genetics, Faculty of Arts and Sciences, Bogazici University, Istanbul, Turkey

Corresponding author
Seyfullah Kotil, enesseyful-lah.kotil@med.bau.edu.tr, enesseyfullah.kotil@boun.edu.tr

## ABSTRACT

Drug resistance is a primary barrier to effective treatments of HIV/AIDS. Calculating quantitative relations between genotype and phenotype observations for each inhibitor with cell-based assays requires time and money-consuming experiments. Machine learning models are good options for tackling these problems by generalizing the available data with suitable linear or nonlinear mappings. The main aim of this study is to construct drug isolate fold (DIF) change-based artificial neural network (ANN) models for estimating the resistance potential of molecules inhibiting the HIV-1 protease (PR) enzyme. Throughout the study, seven of eight protease inhibitors (PIs) have been included in the training set and the remaining ones in the test set. We have obtained 11,803 genotype-phenotype data points for eight PIs from Stanford HIV drug resistance database. Using the leave-one-out (LVO) procedure, eight ANN models have been produced to measure the learning capacity of models from the descriptors of the inhibitors. Mean $R^2$ value of eight ANN models for unseen inhibitors is 0.716, and the 95% confidence interval (CI) is [0.592–0.840]. Predicting the fold change resistance for hundreds of isolates allowed a robust comparison of drug pairs. These eight models have predicted the drug resistance tendencies of each inhibitor pair with the mean 2D correlation coefficient of 0.933 and 95% CI [0.930–0.938]. A classification problem has been created to predict the ordered relationship of the PIs, and the mean accuracy, sensitivity, specificity, and Matthews correlation coefficient (MCC) values are calculated as 0.954, 0.791, 0.791, and 0.688, respectively. Furthermore, we have created an external test dataset consisting of 51 unique known HIV-1 PR inhibitors and 87 genotype-phenotype relations. Our developed ANN model has accuracy and area under the curve (AUC) values of 0.749 and 0.818 to predict the ordered relationships of molecules on the same strain for the external dataset. The

currently derived ANN models can accurately predict the drug resistance tendencies of PI pairs. This observation could help test new inhibitors with various isolates.

## INTRODUCTION

Acquired immunodeficiency syndrome (AIDS) disease caused by the human immunodeficiency viruses, HIV-1 and HIV-2, began to spread in the 1970s and came into focus in the early 1980s as one of the most severe public health threats in history (*Sharp & Hahn, 2011*). Detection of reverse transcription activity in cultures of lymph node cells from AIDS patients in the early 1980s revealed that AIDS was caused by a retrovirus later called human immunodeficiency virus (HIV) (*Das & Arnold, 2013*). Zidovudine (AZT), the first nucleotide reverse transcriptase inhibitor (NRTI) that inhibits the reverse transcription enzyme of HIV, was approved in 1987, and today there are nearly thirty approved drugs (*Lu et al., 2018*). HIV-1 has affected approximately 38 million people today, and just about 26 million people are receiving "highly active antiretroviral treatment" (HAART) (*Jespersen et al., 2021*). The HAART therapy proposed in the mid-1990s was defined as the procedure of using three or four different drugs that act on various targets in the virus's life cycle (*Palmisano & Vella, 2011*). With HAART therapy, the death rate fell to 47% in 1997, just ten years after the first AIDS case was detected (*World Heath Organization*).

Drug resistance is the primary barrier to the effective treatment of HIV/AIDS (*Günthard et al., 2019*; *Kuritzkes, 2011*). Single-drug treatments for HIV yield rapid resistance due to the high genetic diversity and error-prone replication of the virus (*Kuritzkes, 2011*; *Oroz et al., 2019*). Hence, the use of drug combinations through the HAART protocols increases the efficacy of the treatment (*Lagnese & Daar, 2008*). However, cross-resistant isolates for available drugs encourage researchers to find novel inhibitors (*Cihlar et al., 2006*; *Stranix et al., 2003*; *Nakatani et al., 2008*; *Koh et al., 2009*; *Koh et al., 2003*). To combat drug-resistant isolates, novel drug design methodologies have been adopted for HIV-1 protease enzyme such as phosphonate-mediated solvent anchoring (*Cihlar et al., 2006*), lysine sulfonamide-based molecular core (*Stranix et al., 2003*), allophenylnorstatine containing inhibitors (*Nakatani et al., 2008*), nonpeptic inhibitor GRL-02031 (*Koh et al., 2009*), bis-tetrahydrofuranylurethane containing nonpeptidic inhibitor UIC-94017 (*Koh et al., 2003*). Testing novel inhibitors with various drug-resistant isolates need experimental or computational mechanisms.

HIV protease enzyme plays a vital role in forming infectious viruses by regulating immature viruses' synthesized gag and gag-pol polyproteins (*Zhang, Kaplan & Tropsha, 2008*). Protease inhibitors are generally included in the scope of HAART therapy, and eight approved drug molecules are used effectively today (*World Heath Organization*). Dose–response curves of protease inhibitors were shown that they have higher Hill coefficient
values than the fusion (FI), integrase (II), nucleoside reverse transcriptase (NRTI), and non-nucleotide reverse transcriptase (NNRTI) inhibitors (*Rosenbloom, Hill & Rabi, 2012*). Even if a person is infected with the wild-type virion, resistant variants may emerge with dosing disruptions or the use of inappropriate combinations in the HAART therapy (*Jilek et al., 2011*). The success rate of HAART therapy can be increased by measuring the efficacy of existing and novel inhibitors over resistant genotypes (*Xing et al., 2013*; *Lima et al., 2008*). Observing drug-efficacy relations with cell-based assays is expensive and time-consuming in the presence of genotype information. Mathematical models are essential to tackle this important problem (*Wei et al., 2015*; *Yu, Weber & Harrison, 2014*; *Hosseini, Alibés & Noguera-Julian, 2016*).

Various mathematical models have been calibrated using genotype-phenotype change data proposed in the Stanford HIV database to predict mutational effects on viral dynamics in the literature (*Talbot et al., 2010*; *Obermeier et al., 2012*; *Van Laethem et al., 2002*; *Meynard et al., 2002*; *Amamuddy, Bishop & Bishop, 2017*; *Amamuddy, Bishop & Bishop, 2018*; *Wang & Larder, 2003*; *Drăghici & Potter, 2002*; *Kjaer et al., 2008*; *Steiner, Gibson & Crandall, 2020*; *Wang et al., 2009*; *Shen et al., 2016*; *Shah et al., 2020*; *Tarasova et al., 2020*; *Tarasova et al., 2018*; *Ota et al., 2021*). The life span of patients can be considerably extended by constructing reliable mathematical models that accurately predict suitable drugs for existing isolates. Most existing prediction models are knowledge-based and require predetermined rules on mutations and drugs (*Talbot et al., 2010*; *Obermeier et al., 2012*; *Van Laethem et al., 2002*; *Meynard et al., 2002*). The most commonly used genotype interpretation algorithms have been observed to be Stanford HIVdb (*Talbot et al., 2010*), HIV-grade (*Obermeier et al., 2012*), REGA (*Van Laethem et al., 2002*), and ANRS (*Meynard et al., 2002*). In addition to these genotype interpretation algorithms, various machine learning models have recently been proposed to predict genotype-phenotype change relationships in the presence of a predetermined inhibitor (*Amamuddy, Bishop & Bishop, 2017*; *Amamuddy, Bishop & Bishop, 2018*; *Wang & Larder, 2003*; *Drăghici & Potter, 2002*; *Kjaer et al., 2008*; *Steiner, Gibson & Crandall, 2020*; *Wang et al., 2009*; *Shen et al., 2016*; *Shah et al., 2020*; *Tarasova et al., 2020*; *Tarasova et al., 2018*; *Ota et al., 2021*). Artificial neural network (*Amamuddy, Bishop & Bishop, 2017*; *Amamuddy, Bishop & Bishop, 2018*; *Wang & Larder, 2003*; *Drăghici & Potter, 2002*; *Kjaer et al., 2008*; *Steiner, Gibson & Crandall, 2020*), random forest algorithm (*Wang et al., 2009*; *Shen et al., 2016*; *Shah et al., 2020*; *Tarasova et al., 2020*; *Tarasova et al., 2018*; *Ota et al., 2021*; *Cai et al., 2021*), support vector machine (*Shah et al., 2020*; *Cai et al., 2021*; *Beerenwinkel et al., 2003*), decision trees (*Beerenwinkel et al., 2002*), k-nearest neighbors (kNN) (*Shen et al., 2016*), restricted Boltzmann machine (*Pawar et al., 2018*), support vector regression (*Ota et al., 2021*) and linear regression (*Rhee, Taylor & Fessel, 2010*) are the methods used in the literature to model the efficacy of different drugs against HIV-1 variants. All the works mentioned above focus on predicting the fold change of mutant fitness under a single drug. Fold change values for each molecule are treated as disjoint and used to construct a drug-specific model. Those type of models does not need to take molecular descriptors of a drug as input, hence are indifferent to chemical structure. Such models cannot predict

the effects of resistance mutations for a novel drug. Therefore, a model that predicts fold change of mutant fitness for multiple molecules is needed.

Here, a machine learning model was constructed that simultaneously takes molecular fingerprints and mutational information jointly as inputs to estimate the fold change values. For training and testing sets, we used data from eight approved protease inhibitors atazanavir (AZT), darunavir (DRV), fosamprenavir (FPV), indinavir (IDV), lopinavir (LPV), nelfinavir (NFV), saquinavir (SAV) and tipranavir (TPV) in the Stanford HIV drug resistance database. By imposing leave one out (LVO) test procedure, our drug-isolate-fold (DIF) change-based artificial neural network (ANN) models are seen to have the ability to learn both from inhibitor descriptors and mutational genotype information to predict fold-change values. The model can predict the fold change of hundreds of isolates. To that end, the learned hundreds of predictors(fold-change of isolates) can be successfully used to assess the resistance potential of inhibitors. We used pairs of drugs to predict the more resistance-prone molecule. We called these pairwise comparisons the resistance tendencies. Our DIF-based ANN models predicted each protease inhibitor (PI) pair's drug resistance tendencies accurately, and these quantitative results support our central arguments.

## METHODS AND MATERIAL

### Dataset description

Filtered genotype-phenotype data on the Stanford HIV drug resistance database was retrieved for PIs (*Das & Arnold, 2013*). We have organized this dataset with respect to isolates and inhibitors, and 498 protease mutations have been observed. For the HIV-1 PI: 1218 isolates for atazanavir (ATV), 678 isolates for darunavir (DRV), 1809 isolates for fosemprenavir (FPV), 1,860 isolates for indinavir (IDV), 1,562 isolates for lopinavir (LPV), 1907 isolates for nelfinavir (NFV), 1,861 isolates for saquinavir (SQV) and 908 isolates for tipranavir (TPV) have been analyzed for PI susceptibility. In the dataset, 436, 336, 480, 483, 472, 486, 489 and 409 different mutations have been observed for ATV, DRV, FPV, IDV, LPV, NFV, SQV, and TPV, respectively.

### Representation of isolates

Four hundred ninety-eight unique mutations were observed in the eight protease inhibitors dataset. The binary barcoding technique was applied here to represent the isolates that occurred in the dataset, as also used in several studies of modeling genotype-phenotype data for various HIV-1 inhibitors (*Lu et al., 2018*). Thus, a 498-dimensional vector of binary entries with 0s and 1s uniquely representing any existing isolates is considered. Assume that the 498 unique mutations produce the vector $X = [x_1, x_2, \ldots, x_{498}]$ where $x_i$ is a mutation pattern that occurred in the dataset. For instance, $x_1$ denotes the occurrence of the mutation A22S or $x_{478}$ denotes the occurrence of the mutation V82A. For example, the isolate $I_j = [\text{A22S}, \text{V82A}]$ can be barcoded as $X = [1, 0, \ldots, 0, 1, 0, \ldots, 0]$ in which only the first and four hundred seventy-eighth position take value one, and the remaining entries have value zero. Any isolate can be obtained from any combination of these mutations, and the isolate $j$ can be defined as $I_j = \{a_1, a_2, \ldots, a_n\}$ with

$$a_k = \begin{cases} 1, & if \ x_k \in I_j \\ 0, & otherwise. \end{cases}$$

In this way, each isolate can be transformed into a unique 498-dimensional input vector used for the machine learning. The binary barcoding approach has the advantage of representing two or more mutational changes in the amino acids since each mutation has a unique position in the 498-dimensional input vector. For example, assuming that $x_{480}$ and $x_{481}$ denotes mutations V82F and V82I, the isolate $I = [V82F, V82I]$ can be represented as $X = [0, 0, \ldots, 0, 1, 1, 0, \ldots, 0]$. It is important to note that, the genotype-fold change measurements are made on population of viruses, so that it is possible to find two separate mutations for a single residue.

## Representation of inhibitors

To construct a drug-isolate-fold change model for the HIV-1 protease inhibitors, the molecular representations of the inhibitors have been built with binary Morgan fingerprints. The Morgan fingerprints provide an effective way of the vector representations of molecules and are widely used in machine learning models (*Jespersen et al., 2021*). The RDKit environment of the Python program has been used to convert the smile representations of ATV, DRV, FPV, IDV, LPV, NFV, SQV, and TPV inhibitors to a binary 512-bit vector representation. 234 out of 512 bits have been seen to provide unique characteristics for 8 PI. Thus, the molecular representation of each PI needs 234-dimensional vectors.

## Artificial neural network (ANN) model for regression

An ANN model has been constructed with isolate-inhibitor inputs and fold change outputs with the machine learning and deep learning toolbox of the MATLAB program. Since isolates and inhibitors are uniquely represented by 498- and 234- dimensional vectors, the ANN model has 732-dimensional input. The ANN architecture includes 732-dimensional input, five hidden layer neurons, and one output neuron with hyperbolic tangent-sigmoid and linear activation functions. Logarithms of fold-change values in the dataset are taken as output variables of the neural network models. In the training process, the scaled conjugate gradient algorithm with MATLAB built-in function "trainscg" is utilized over GPU (*Palmisano & Vella, 2011*).

## Ensemble processing

Since we have only eight inhibitors, measuring the molecular learning capacity of our ANN model is crucial. In this way, an ensemble learning procedure is used to improve the molecular learning performance of the model. For each PI, the $100 \times 50$ model has been trained with the data of the remaining seven inhibitors. From every 50 models, a model is chosen that yields the minimum mean square error for the interior test set of the corresponding PI data. Thus, 100 optimal models are obtained, and the final model is calculated as the average of these models.
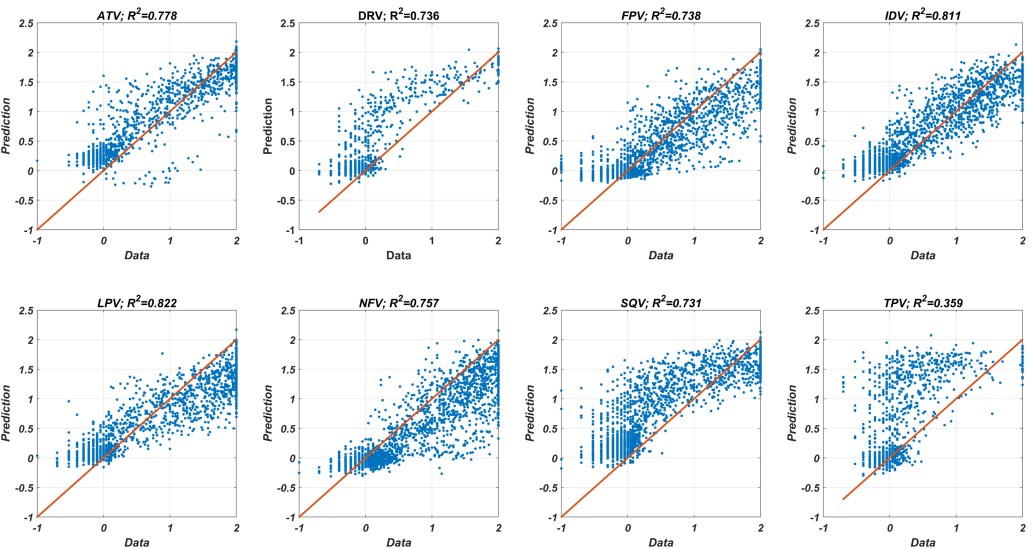

**Figure 1** **Data *versus* the predicted fold change values from DIF-based ANN models.** DIF-based ANN regression models are constructed with the LVO testing methodology. For each figure, the fold-change results are estimated by an ANN model, which is trained with the remaining data of the seven PIs. The $R^2$ values correspond to the square of the linear correlation coefficient of the data and prediction.

# RESULTS

## Regression performance of molecular learning models

Eight feed-forward neural network models have been constructed with drug-isolate-fold-change (DIF) data by excluding one of the drugs from training in each case. The ANN model was trained with the remaining seven DIF data predicted the excluded results. The sizes of the training and test sets were changed according to the excluded PIs (mean values are 10,328 and 1,475 for training and test sets, respectively). The regression performances of each model are illustrated in Fig. 1 with corresponding $R^2$ values (square of the linear correlation coefficient). The best and worst results are obtained by predicting the outcomes of the drugs LPV and TPV with $R^2 = 0.822$ and $R^2 = 0.359$, respectively. Similarly, predicting the fold-change results of the inhibitor TPV was observed to be the worst in the literature (*Yu, Weber & Harrison, 2014*). The mean $R^2$ value of all predictions is 0.716 and the 95% confidence interval is [0.592–0.840]. The DIF-based ANN model provides accurate estimations even if the test data consists of unseen drugs. This observation implies that our ANN models accurately learn molecular information from the Morgan fingerprints. The detailed performance results of our DIF-based ANN models are presented in Table 1.

An inevitable question is how molecular information changes the regression and classification performance of our ANN models. To clarify this, we trained isolate-fold-change (IF) based ANN models for each inhibitor and compared them with the current DIF-based models. In Figs. S1–S2, the model performances have been compared by measuring $R^2$ and area under the curve (AUC) values for each PI. Table S1 also shows the accuracy, sensitivity, specificity, and Matthews correlation coefficient (MCC) scores

**Table 1  Mean square error (MSE) and $R^2$ values of the DIF-based ANN models for predicting the logarithmic fold change values of 8 PIs are presented[b].**

| ARVs[a] | $R^2$ | | | MSE | |
|---|---|---|---|---|---|
| | Whole dataset | Test set | | Whole dataset | Test set |
| ATV | 0.865 | 0.778 | | 0.087 | 0.166 |
| DRV | 0.857 | 0.736 | | 0.092 | 0.227 |
| FPV | 0.849 | 0.738 | | 0.097 | 0.160 |
| IDV | 0.861 | 0.811 | | 0.090 | 0.131 |
| LPV | 0.852 | 0.822 | | 0.096 | 0.188 |
| NFV | 0.845 | 0.757 | | 0.101 | 0.215 |
| SQV | 0.833 | 0.731 | | 0.109 | 0.283 |
| TPV | 0.821 | 0.359 | | 0.116 | 0.560 |

Notes.

[a]Abbreviations: ATV, atazanavir; DRV, darunavir; FPV, fosamprenavir; IDV, indinavir; LPV, lopinavir; NFV, nelfinavir; SQV, saquinavir; TPV, tipranavir.

[b]Drug-isolate-fold change models are constructed as a general neural network model taking drug fingerprints and mutation information as inputs. For each row, the corresponding drug has not been included in the training process. The test set performance of each model has been evaluated with respect to the excluded drugs. $100 \times 50$ simulations with random weights have been done, and 100 neural network models that yield minimum MSE for interior test set among 50 trials are obtained. The final neural network model is achieved by taking the mean of 100 models.

of both DIF-based and IF-based models. These findings suggest that the DIF-based ANN model performs slightly better in terms of regression and classification for six of the eight PIs. The two models are compatible with the remaining two inhibitors (DRV and TPV). Therefore, the molecular information used by the DIF-based model provides a better predictive capability. It is very important to note that IF-based models can never predict fold-change values for novel molecules. The real distinction between DIF and IF-based models is that the DIF-based model can predict fold-change values for a novel drug. Hence, the DIF-based ANN model has both better learning capability from mutant information (by comparing to the IF-based model) and the ability to test novel molecules for a given isolate.

### Prediction of drug resistance tendencies for each PI pair

The inhibition potential of each PI in the presence of various genotypes is known to be variable. Tendencies of the logarithmic fold change values for each PI pair provide valuable information about the resistance profiles of the inhibitors, as seen in Fig. 2. Prediction of these tendencies by the DIF-based ANN models and the corresponding 2D correlation coefficients are presented in Fig. 2 in a comparative way for each PI pair. For each PI, prediction has been made with the ANN model trained by the data of the remaining seven inhibitors with an ensemble learning approach. This procedure shows the molecular learning capacity of our ANN models from the Morgan fingerprints. The minimum and maximum 2D correlation coefficients are 0.892 and 0.954 for TPV-DRV and LPV-DRV couples (95% CI [0.930, 0.938]), respectively. Thus, the current DIF-based ANN models can distinguish the inhibitory potentials of each PI pair.
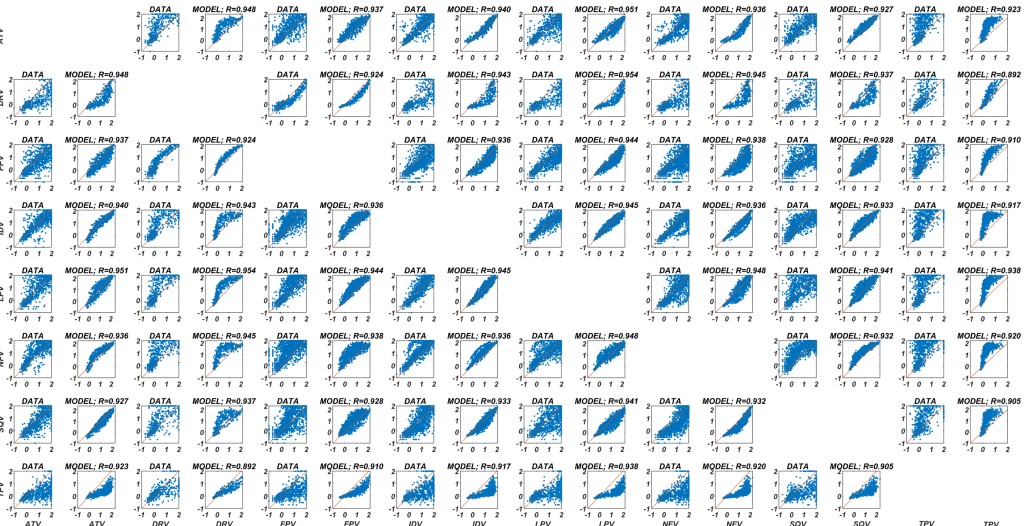

**Figure 2** **Prediction of the fold-change tendencies with the DIF-based ANN model for each PI pair.** The common isolate data of each PI pair and the corresponding DIF-based ANN model predictions are illustrated with 2D correlation coefficients. For each PI, the prediction is constructed using the DIF-based ANN model, which is trained with the remaining seven PI data. The illustrations show the tendencies of the drug resistances for each PI pair for common genotypes.

## Classification of PIs with respect to possible common isolates

Our DIF-based ANN models can distinguish the fold change values of each PI in the presence of any isolate. In this way, a classification problem measuring the relationship $log(FoldChange[A, Isolate]) > log(FoldChange[B, Isolate])$ has been constructed, where A and B are possible protease inhibitors. These relations take values 0 and 1 depending on the inhibitors and isolates. Therefore, our ANN models have been trained with the data from seven inhibitors, with the exception of one particular inhibitor considered as test data. The corresponding receiver operating characteristic (ROC) curves are illustrated in Fig. 3. Area under the ROC curve (AUC) values are included in the figure. The best and worst AUC values have been obtained for the IDV-LPV and DRV-LPV pairs with 0.992 and 0.818 (95% CI: [0.950–0.978]), respectively. In this context, the current DIF-based ANN models have ability to capture the binary relations between any PI pair with high approximation performance.

Performance metrics of the current ANN models for capturing binary relations of PI pairs are presented in Table 2. As indicated in the table, the DIF-based ANN models have a high rate of true prediction for each PI pair. The mean accuracy, sensitivity, specificity and Matthews correlation coefficient (MCC) values have been computed as 0.954, 0.791, 0.791 and 0.688 (95% CI [0.932–0.952], [0.719–0.863], [0.719–0.863] and [0.600–0.776]), respectively. The most conspicuous result here is that the neural network models can classify the inhibitors for resistance profiles, even if that model did not see the corresponding inhibitors in the training process.

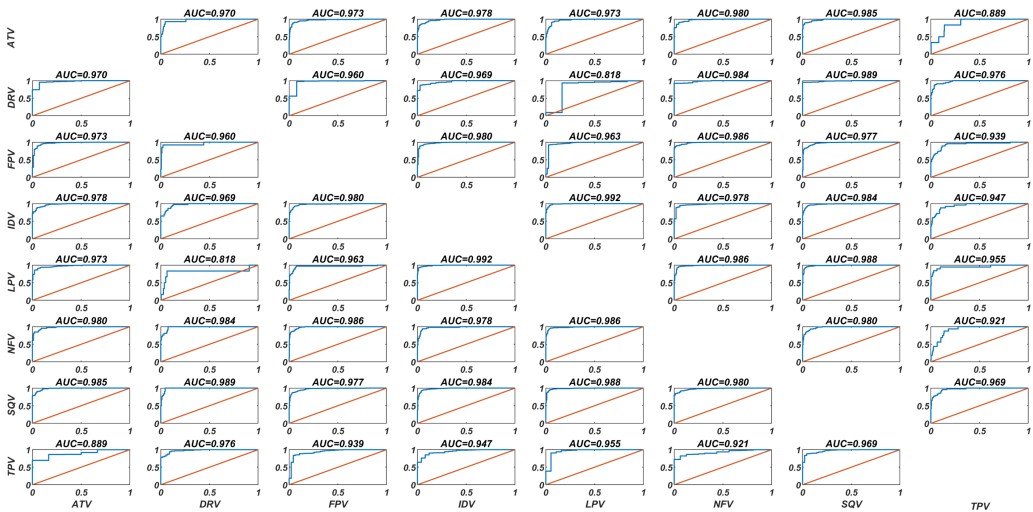

**Figure 3** **Classification performances of the DIF-based ANN models.** The DIF-based ANN classification models are constructed with the LVO methodology. For each PI, the classification of resistant and non-resistant isolates is estimated by an ANN model trained with the remaining data of the seven PI. The AUC values correspond to the area under the ROC curves, and the accuracy is evaluated with the true estimation rate.

## Testing the molecular learning model with the external data set

For the external dataset, we conducted a search for compounds that were comparable to the eight HIV PIs that were already available in the ChEMBL database (*Sevenich et al., 2017*). An initial set of 1,305 compounds and their biological activity values were extracted. First, compounds having 70% or less similarity to each drug were filtered out. Then, molecules with determined $IC_{50}$ values in mutant viruses were collected. The compounds with determined $IC_{50}$ values in mutant viruses were then gathered. Furthermore, the maximum Tanimoto similarities of these molecules to the current eight PIs were computed (using 512-bit Morgan fingerprints), and molecules with 40% or less structural similarity were filtered out. Finally, molecules with 1s in the discarded 278-dimensional fingerprint vectors, that is, molecules with information loss, were filtered out. A final dataset of 87 genotype-phenotype relationships involving 51 different molecules was obtained (see Data S1). Only six of these molecules are in the existing PI set (DRV, IDV, NFV, ATV, LPV) and only 20 of 87 genotype-phenotype relations belong to these six PIs. In the presence of eight distinct strains, there is a fold change difference in $IC_{50}$ values for these molecules. Fortunately, the data includes different molecules tested on the same protein, so we can evaluate the efficiency of our ANN model on ranking of these molecules.

We have used the eight existing PIs (see Fig. 4 for chemical structures) to construct an ANN model as described in the *Methods and Material* section. The main difference is the consideration of all molecules rather than the LVO procedure. The Morgan fingerprints of the new 51 molecules were determined, and then the similar mapping was used to reduce the 512-dimensional vectors into 234-dimensional inputs. It should be noted that the molecules were chosen in such a way that there are no 1s in the discarded 278 bits. This

**Table 2  Accuracy, sensitivity, specificity and MCC values of the DIF-based ANN models for predicting the drug resistance tendencies for each couple of ARVs[a].**

| ARVs | | ATV | DRV | FPV | IDV | LPV | NFV | SQV | TPV |
|---|---|---|---|---|---|---|---|---|---|
| ATV | Accuracy | – | 0.970 (224/231) | 0.932 (438/470) | 0.923 (264/286) | 0.913 (303/332) | 0.948 (361/381) | 0.896 (301/336) | 0.983 (404/411) |
| | Sensitivity | – | 0.500 (7/14) | 0.772 (78/101) | 0.850 (85/100) | 0.978 (178/182) | 0.986 (291/295) | 0.720 (90/125) | 0.000 (0/6) |
| | Specificity | – | 1.000 (217/217) | 0.976 (360/369) | 0.962 (179/186) | 0.833 (125/150) | 0.814 (70/86) | 1.000 (211/211) | 0.998 (404/405) |
| | MCC | – | 0.696 | 0.791 | 0.829 | 0.828 | 0.856 | 0.786 | 0.000 |
| DRV | Accuracy | 0.970 (224/231) | – | 0.968 (184/190) | 0.917 (222/242) | 0.960 (215/224) | 0.976 (321/329) | 0.936 (206/220) | 0.897 (156/174) |
| | Sensitivity | 1.000 (217/217) | – | 0.972 (172/177) | 0.966 (198/205) | 0.982 (214/218) | 1.000 (308/308) | 0.972 (173/178) | 0.714 (40/56) |
| | Specificity | 0.500 (7/14) | – | 0.923 (12/13) | 0.649 (24/37) | 0.167 (1/6) | 0.619 (13/21) | 0.786 (33/42) | 0.983 (116/118) |
| | MCC | 0.696 | – | 0.792 | 0.662 | 0.162 | 0.777 | 0.788 | 0.761 |
| FPV | Accuracy | 0.932 (438/470) | 0.968 (184/190) | – | 0.936 (677/723) | 0.952 (511/537) | 0.964 (878/911) | 0.930 (705/758) | 0.932 (369/396) |
| | Sensitivity | 0.976 (360/369) | 0.923 (12/13) | – | 0.993 (552/556) | 0.996 (465/467) | 0.996 (817/820) | 0.975 (502/515) | 0.511 (24/47) |
| | Specificity | 0.772 (78/101) | 0.972 (172/177) | – | 0.749 (125/167) | 0.657 (46/70) | 0.670 (61/91) | 0.835 (203/243) | 0.989 (345/349) |
| | MCC | 0.791 | 0.792 | – | 0.816 | 0.771 | 0.782 | 0.838 | 0.630 |
| IDV | Accuracy | 0.923 (264/286) | 0.917 (222/242) | 0.936 (677/723) | – | 0.952 (399/419) | 0.952 (498/523) | 0.929 (562/605) | 0.957 (404/422) |
| | Sensitivity | 0.962 (179/186) | 0.649 (24/37) | 0.749 (125/167) | – | 0.989 (270/273) | 0.994 (468/471) | 0.874 (221/253) | 0.280 (7/25) |
| | Specificity | 0.850 (85/100) | 0.966 (198/205) | 0.993 (552/556) | – | 0.884 (129/146) | 0.577 (30/52) | 0.969 (341/352) | 1.000 (397/397) |
| | MCC | 0.829 | 0.662 | 0.816 | – | 0.895 | 0.702 | 0.854 | 0.518 |
| LPV | Accuracy | 0.913 (303/332) | 0.960 (215/224) | 0.952 (511/537) | 0.952 (399/419) | – | 0.944 (526/557) | 0.929 (509/548) | 0.982 (429/437) |
| | Sensitivity | 0.833 (125/150) | 0.167 (1/6) | 0.657 (46/70) | 0.884 (129/146) | – | 0.979 (375/383) | 0.836 (173/207) | 0.632 (12/19) |
| | Specificity | 0.978 (178/182) | 0.982 (214/218) | 0.996 (465/467) | 0.989 (270/273) | – | 0.868 (151/174) | 0.985 (336/341) | 0.998 (417/418) |
| | MCC | 0.828 | 0.162 | 0.770 | 0.895 | – | 0.869 | 0.850 | 0.755 |
| NFV | Accuracy | 0.948 (361/381) | 0.976 (321/329) | 0.964 (878/911) | 0.952 (498/523) | 0.944 (526/557) | – | 0.935 (735/786) | 0.966 (477/494) |
| | Sensitivity | 0.814 (70/86) | 0.619 (13/21) | 0.670 (61/91) | 0.577 (30/52) | 0.868 (151/174) | – | 0.451 (41/91) | 0.188 (3/16) |
| | Specificity | 0.986 (291/295) | 1.000 (308/308) | 0.996 (817/820) | 0.994 (468/471) | 0.979 (375/383) | – | 0.999 (694/695) | 0.992 (474/478) |
| | MCC | 0.846 | 0.777 | 0.782 | 0.702 | 0.869 | – | 0.639 | 0.268 |
| SQV | Accuracy | 0.896 (301/336) | 0.936 (206/220) | 0.930 (705/758) | 0.929 (562/605) | 0.929 (509/548) | 0.935 (735/786) | – | 0.898 (359/400) |
**Table 2** (*continued*)

| ARVs | | ATV | DRV | FPV | IDV | LPV | NFV | SQV | TPV |
|------|--|-----|-----|-----|-----|-----|-----|-----|-----|
| SQV | Sensitivity | 1.000 (211/211) | 0.786 (33/42) | 0.835 (203/243) | 0.969 (341/352) | 0.985 (336/341) | 0.999 (694/695) | – | 0.146 (7/48) |
| | Specificity | 0.720 (90/125) | 0.972 (173/178) | 0.975 (502/515) | 0.874 (221/253) | 0.836 (173/207) | 0.451 (41/91) | – | 1.000 (352/352) |
| | MCC | 0.786 | 0.788 | 0.838 | 0.854 | 0.850 | 0.639 | – | 0.361 |
| TPV | Accuracy | 0.983 (404/411) | 0.897 (156/174) | 0.932 (369/396) | 0.957 (404/422) | 0.982 (429/437) | 0.966 (477/494) | 0.898 (359/400) | – |
| | Sensitivity | 0.998 (404/405) | 0.983 (116/118) | 0.989 (345/349) | 1.000 (397/397) | 0.998 (417/418) | 0.992 (474/478) | 1.000 (352/352) | – |
| | Specificity | 0.000 (0/6) | 0.714 (40/56) | 0.511 (24/47) | 0.280 (7/25) | 0.632 (12/19) | 0.188 (3/16) | 0.146 (7/48) | – |
| | MCC | 0.000 | 0.761 | 0.630 | 0.518 | 0.755 | 0.268 | 0.361 | – |

**Notes.**

[a]Accuracy, sensitivity and specificity values represent the rate of true predictions, true positive rate and true negative rate, respectively. The common genotype data is used for each PI pair by eliminating the observations satisfying $\left|\log(A) - \log(B)\right| \leq \log 2$ where $A$ and $B$ are the fold change values of drugs A and B for a specified genotype.

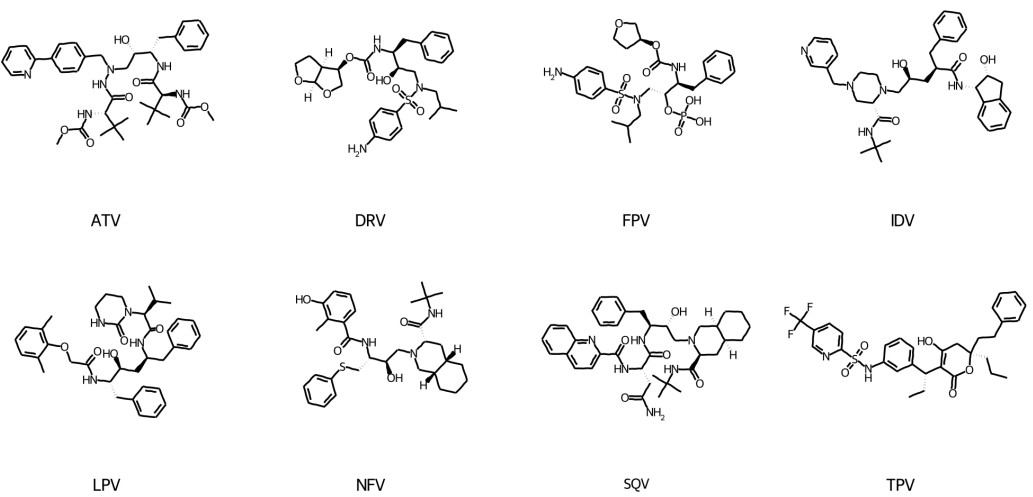

ATV     DRV     FPV     IDV

LPV     NFV     SQV     TPV

**Figure 4**   **Chemical structures of eight PIs utilized in the DIF-based ANN model training.**

condition was specifically designed into the external dataset. In this approach, we ensure that no meaningful information for novel molecules is lost during the machine learning process.

Two different classification performances of our ANN model were observed to test its molecular learning potential. The first is the classification of resistant and susceptible strains for a given PI, with a threshold fold chance value of 3 (*Shen et al., 2016*). The goal of this classification task is to evaluate the resistance labeling performance of the current model for diverse molecules. As demonstrated in Table 3, the accuracy, sensitivity, specificity, MCC, and AUC metrics of our model for labeling the resistance in external data are 0.678, 0.536,

**Table 3  Classification performance of the ANN model on the external test data.** In resistance classification procedure, the ANN model is used to classify the genotypes as drug resistant (Fold Change ≥3) and susceptible (Fold Change <3) for each external test data. On the other hand, for ranking classification procedure, the ANN ranks the drugs for a given genotype in terms of their logarithmic fold change values.

| Model / Metric | Accuracy | Sensitivity | Specificity | MCC | AUC |
|---|---|---|---|---|---|
| Resistance Classification | 0.678 | 0.536 | 0.935 | 0.468 | 0.843 |
| Ranking Classification | 0.749 | 0.767 | 0.730 | 0.497 | 0.819 |

**ChEMBL ID** = CHEMBL405134
**Experimental Fold-change** = 436.52
**Predicted Fold-change** = 2.82

**ChEMBL ID** = CHEMBL115
**Experimental Fold-change** = 1.15
**Predicted Fold-change** = 1.74

**Figure 5  Resistance classification for two compounds having similar structures.** A ranking classification example for two compounds that have similar structures with different resistance profiles for the same strain (L10I, L19Q, K20R, E35D, M36I, S37N, M46I, I50V, I54V, I62V, L63P, A71V, V82A, L90M, E28K, K32E, V35I, T39T, E40DV, M41L, K43E, Y181Y).

0.935, 0.468, and 0.843, respectively. Our ANN model performs satisfactorily statistically on the external dataset as a result.

Our major point is that the built ANN model is capable of ranking the efficiency of PIs for a given strain. The external dataset contains 853 pair of resistance scores for 51 different molecules. Our ANN model predicted these 853 pair of resistance scores as well, and we measured the ranking performance of the model. This testing approach is identical to the previously described tendency and ranking assessments of the eight existing PIs. For this classification task, the accuracy, sensitivity, specificity, MCC, and AUC scores have been 0.749, 0.767, 0.730, 0.497, and 0.819, respectively (see Table 3). As a result, our ANN model appropriately ranks the compounds based on their resistance profiles (see Fig. 5 for a representative example). The ROCs for both resistance classification and ranking classification performances of the ANN model can be seen in Fig. 6. The ANN model learned considerable information from the molecular structure of the eight PIs, according to the ROCs. In summary, if the external PIs have no information loss (no 1s in the discarded 278 bits within 512 bits) in comparison to the existing eight PIs, our ANN model has a high ability to rank these PIs.

One method for reducing and visualizing high-dimensional data is principal component analysis (PCA). It is widely used to describe the chemical space occupied by a set of

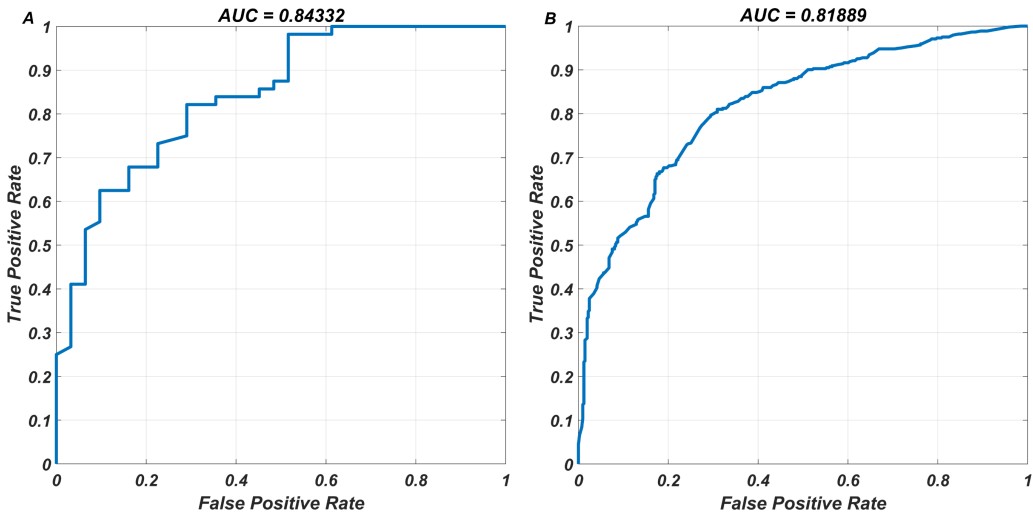

**Figure 6** **The ROCs corresponding to resistance and ranking classifications for the test data.** The DIF-based ANN model has been used to (A) classify the resistance and susceptible strains (B) classify the rankings of 853 pair of resistance scores, for various molecules existing our external data. The AUC ratings associated with the ROC curves measure how threshold probabilities affect FPR-TPR pairs in classification tasks.

molecules. When the descriptors of the compounds are used for analysis, it allows for the clustering of similar molecules as well as the distinguishing of diverse molecules. To verify the applicability of our model in terms of chemical similarity and diversity, we have performed the PCA on the external set molecules using 234-dimensional vectors employed for fold-change prediction. We only evaluated the unique molecules in the external set, and for molecules that were tested in several strains and had multiple fold-change prediction values, we have taken the values with the largest absolute prediction error (AE). As the first two components, PC1 and PC2, were able to represent more than half of the variance in the data and accurately depict the correlations between the similarities of the molecules, 234-dimensional vectors for each unique molecule were projected to 2D-space. Figure 7 displays the PCA plot for both training and external set compounds and colored according to the AE for fold-change prediction or training compound name. According to the figure, other training compounds other than the IDV were not involved in cluster formation with external set molecules. Though the same external set molecules were not forming clusters, still there were three clusters that contain three or more molecules. For each of these cluster, we have selected one representative structure and additionally we have found the maximum common substructure (MCS) using the FMCS algorithm implemented in RDKit. We discovered that for all compounds in the cluster with the representative structure A in Fig. 7, the fold-changes were predicted with low errors (AE <1.0). The compounds in this cluster have certain substructures with HIV inhibitors, such as a sulfonamide group, a bis-THF alcohol moiety, and a benzodioxole group (*Sevenich et al., 2017*). The only structural difference in this cluster of compounds was observed on the side chain connecting to the phenoxy group. Our model, on the other hand, did
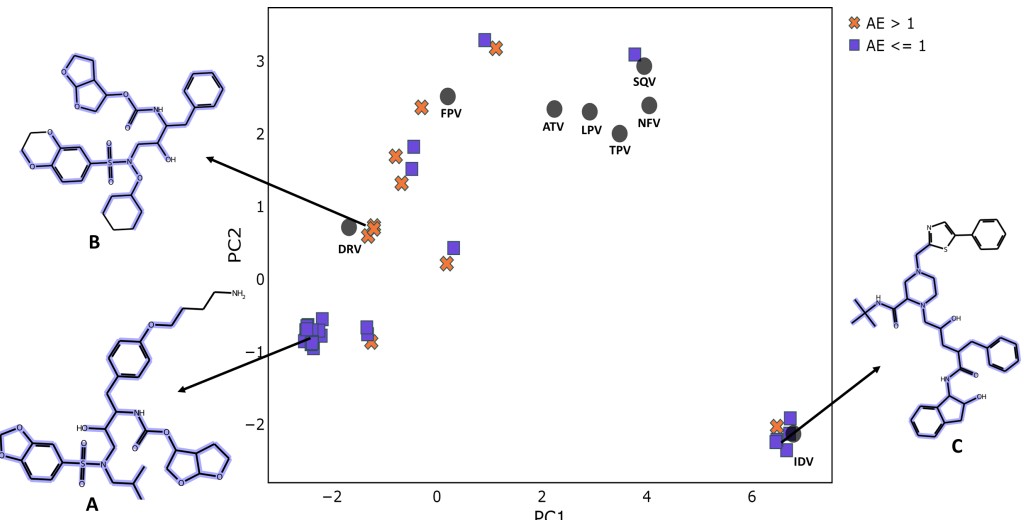

**Figure 7** **PCA plot for external set molecules obtained using the unique characteristic 234 bits fingerprint descriptors.** PCA plot for external set molecules produced with the unique 234-bit fingerprint descriptors. The absolute error (AE) for fold change prediction using the model is calculated for each molecule. Molecules are classified with respect to corresponding AE values (AE =1 is selected as threshold). The data points in black are for the existing eight PIs, which have their names indicated. As demonstrated, example structures from clusters are exhibited and designated A, B, and C. In the 2D depiction, the most prevalent substructures in the same clusters have been highlighted in dark blue.

not perform well for another cluster of compounds represented by structure B in Fig. 7. Despite the fact that the MCS for three compounds in this cluster were similar to the preceding cluster, represented by structure A, there were subtle differences that caused compounds to be in distinct clusters. For these, the cyclohexyl hydroxy or cyclopentyl hydroxy group was attached to the N of the sulfonamide group. Furthermore, the varying side group of the phenoxy group in the first cluster was shorter in the second cluster since there was only phenyl group. Compounds having structure C are represented in another cluster, as seen in Fig. 7. These compounds had a high resemblance to the IDV (see Fig. 4 for a 2D representation of the IDV), and the fold-change was predicted with minimal errors (AE <1.0). These compounds, like the IDV, have substructure groups such as piperazinecarboxamide, indenol derivative, and phenyl group. However, the compounds in this cluster have different attachment groups connected to piperazine ring instead of pyridine group in the IDV. We have also discovered that for the compound where the fold change estimation error above the threshold (AE ≥ 1.0), the exact value was 1.01, indicating that it is negligible. It should be noted, however, that this compound contains a Fluorine substituent, which is known to cause a significant increase in the inhibitory activity of compounds (*Amano et al., 2022*) and it may not be easily learnt using our model. This chemical similarity analysis revealed that our model can predict the fold change for similar compounds with low error, despite the fact that some of the external set molecules based on our descriptors were not similar (or in close proximity in the PCA plot in Fig. 7) to internal set molecules.

# DISCUSSION

This article presents a machine learning approach for predicting fold-change values using HIV-1 protease inhibitor and isolate characteristics. The filtered PhenoSense assay results made accessible in the Stanford HIV drug resistance database have been used training and testing machine learning models. Seven of the eight inhibitors have been used to train drug-isolate-fold change-based feed-forward artificial neural networks, with the remaining one serving as test data (LVO). In this context, the LVO procedure produces an objective testing approach for determining the learning capacity of models from the descriptors of the inhibitors. Both inhibitors and isolates have been encoded using binary mappings, which have been shown to be computationally effective. Because of their acknowledged advantages in molecular machine learning models, the Morgan fingerprints have been exploited as binary mappings of protease inhibitors (*Cai et al., 2021*; *Beerenwinkel et al., 2003*; *Beerenwinkel et al., 2002*). An efficient ensemble process has been proposed and verified through various quantitative experiments to handle the overfitting trouble.

The most significant contribution of this research is the construction of drug-isolate-fold change (DIF)-based ANN models, as opposed to the widely studied isolate-fold change (IF)-based models (*Amamuddy, Bishop & Bishop, 2017*; *Amamuddy, Bishop & Bishop, 2018*; *Wang & Larder, 2003*; *Drăghici & Potter, 2002*; *Kjaer et al., 2008*; *Steiner, Gibson & Crandall, 2020*; *Wang et al., 2009*; *Shen et al., 2016*; *Shah et al., 2020*; *Tarasova et al., 2020*; *Tarasova et al., 2018*; *Ota et al., 2021*). Because the IF models do not take the molecular fingerprints as input, they are insensitive to molecular structure. This study shows the possibility of achieving such a generalized model by feeding models with adequate data from various PIs in the presence of isolates. With the use of the LVO procedure throughout our investigation, the current DIF-based models have been shown to be capable of predicting the drug resistance profiles of unknown inhibitors. Even though the Stanford HIV database only has eight available inhibitors, having many isolates for each inhibitor has contributed to the learning process, and reasonable predictions have been found in the regression performance of remaining inhibitors.

The prediction of drug resistance tendencies for each PI pair is an unavoidable expectation from our DIF-based ANN models. Our generalized models can predict resistance trends with high 2D correlation scores, as demonstrated here. The DIF-based models have provided satisfactory accuracy, sensitivity, specificity, and MCC values by creating classification problems from the tendency relations of each PI pair. The DIF-based ANN model utilizes valuable information from the Morgan fingerprints to predict the fold change values of hidden inhibitors, according to our all-quantitative observations.

We have shown that our ANN model can categorize resistant and susceptible strains as well as rank inhibitors based on resistance profiles for an external dataset. The external dataset is designed in such a way that the unique molecules are comparable enough to any of the primary eight PI and there is no bit loss in the reduction procedure of the Morgan fingerprints from 512 to 234 dimensions. Thus, whenever a molecule satisfies these conditions, our DIF-based model has ability to compare this molecule with existing PIs in terms of their resistance scores.

Instead of building independent separate models for each inhibitor, this study offers a fresh viewpoint on the field by incorporating inhibitor characteristics on the input side of machine learning models. The most obvious drawback of our model is the dearth of protease inhibitors with sufficient genotype-phenotype information. Nevertheless, our encouraging findings have demonstrated that including genotype-phenotype information of novel protease inhibitors will help build more generic drug-isolate-fold change-based machine learning models. Additionally, feeding the DIF-based models with data from many conventional and nonconventional inhibitors may result in a unified model for forecasting drug resistance tendencies for any PI pair in the presence of known genotypes. The drug development process for evolvable diseases, such as HIV, bacterial infections, and cancer should be fundamentally different from diseases such as blood-pressure regulators. A drug needs to be effective and stay effective through the test of evolution. Predicting resistance potentials for drugs is becoming a necessity.

## CONCLUSIONS

This study has revealed the advantages of developing DIF-based models to predict drug resistance profiles. Instead of IF-based models, the current approach has allowed us to investigate a new model that can predict the drug resistance tendencies of PI pairs. Even with only eight PIs available, internal and external test results show that the DIF-based model takes significant information from inhibitor descriptors and leads to satisfactory regression performance. As a result, after finishing this study, it is highlighted on the research agenda to train ANN models with more inhibitors by expanding the existing dataset. In this context, it will be feasible to track the drug resistance profiles of any novel protease inhibitor, and it is strongly believed that these insightful forecasts will be a right direction for moving forward.

### Funding

This work was supported by TUBITAK, 2232–International Fellowship for Outstanding Researchers, Project number 118C244. All the results are in sole responsibility of the authors. The funders had no role in study design, data collection and analysis, decision to publish, or preparation of the manuscript.

### Grant Disclosures

The following grant information was disclosed by the authors:
TUBITAK, 2232–International Fellowship for Outstanding Researchers: 118C244.

### Competing Interests

The authors declare there are no competing interests.

## Author Contributions

- Huseyin Tunc conceived and designed the experiments, performed the experiments, analyzed the data, prepared figures and/or tables, authored or reviewed drafts of the article, and approved the final draft.
- Berna Dogan conceived and designed the experiments, performed the experiments, analyzed the data, prepared figures and/or tables, authored or reviewed drafts of the article, and approved the final draft.
- Büşra Nur Darendeli Kiraz analyzed the data, prepared figures and/or tables, and approved the final draft.
- Murat Sari conceived and designed the experiments, authored or reviewed drafts of the article, and approved the final draft.
- Serdar Durdagi conceived and designed the experiments, authored or reviewed drafts of the article, and approved the final draft.
- Seyfullah Kotil conceived and designed the experiments, prepared figures and/or tables, authored or reviewed drafts of the article, and approved the final draft.

## Data Availability

All data and necessary codes are available at Github and Zenodo:

https://github.com/tnchsyn/hivdrugisolatefoldchange_model; tnchsyn. (2023). tnchsyn/hivdrugisolatefoldchange_model: v1.0 (v1.0). Zenodo. https://doi.org/10.5281/zenodo.7527918.

## Supplemental Information

Supplemental information for this article can be found online at http://dx.doi.org/10.7717/peerj.14987#supplemental-information.

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
