# Peer review of "Prediction of HIV-1 protease resistance using genotypic, phenotypic, and molecular information with artificial neural networks"

_PeerJ, doi:10.7717/peerj.14987_

## Round 0.1 · original submission · Minor Revisions

Our reviewers have some suggestions for the improvement of your manuscript.

Reviewer 1 ·

Basic reporting

The manuscript shows the application of the neural network model to predict the resistance of HIV protease inhibitors against different genotypes of the virus. It is clearly presented, with figures and tables providing evidence of the claims. The novelty of the approach is backed up by the state-of-the-art papers presented by the references cited throughout the text.

Experimental design

The method is briefly described, but the model and raw data are provided on a GitHub page, which other researchers can access to evaluate and access the findings. According to the journal's scope, the study can be classified as a biological application. The research question that the author tackled fills a gap in the scientific knowledge of HIV protease inhibitors, taking into account the high level of mutations for HIV.

Validity of the findings

The model's novelty resides in the analyses of pairs of drug resistance data and the development of a model that can identify potential cross-resistance profiles for unknown chemicals. The model is robust, with internal and external datasets used to validate it. The conclusion offers the significant findings discussed in the manuscript.

Additional comments

The language should be edited to remove some problems scattered along the manuscript, like:
line 242 ...threshold value of fold chance value is selected...
line 272 ... PC1 and PC2 as were able to represent...
line 278 Though same of the external set molecules were not forming clusters, still there were three clusters that contain three or more molecules.
line 296 ...connected to piperazine ring such instead of pyridine...
This is not meant to be an exhaustive list.

Reviewer 2 ·

Basic reporting

The manuscript is well prepared, in general.
However I have some comments and suggestions.
First I suggest that the authors double check English, ideally, with a fluent English speaker.
Second, I ask the authors to change some terms that they used in the manuscript:
- line 87: "genotype-fold" -> "genotype-phenotype"
-

Experimental design

1.Section: "Representation of Isolates"
I suggest that the authors provide detailed description of the binary barcoding technique, probably along with the example of real transformation of a sequence to the set of binary vector. What exactly are x1, x2, x3 etc in the set X = {x1, x2, x3...xn}. In other words, are there any differences between particular changes of an amino acid in each position with a mutation (M184V or M184I). Are specific positions of mutations are taken into account in x1, x2, x3... values?

2. Does obtaining prediction for a pair of protease inhibitors require usage chemical structure descriptors (Morgan fingerprints) for a pair of molecules? If yes then in which way were they generated?

3. I suggest that the authors provide the accuracy of classification of strains into resistant and susceptible based on sequence fingerprints only (i.e. excluding chemical structure fingerprints) and compare the accuracy of prediction for (1) combined descriptors and (2) sequence descriptors only.

Validity of the findings

In general, there are no significant violations in the findings.

---

## Round 0.2 · accepted · Accept

Our reviewers are satisfied with the changes you introduced. I am glad to accept this manuscript for publication.

Reviewer 1 ·

Basic reporting

The revised version of the manuscript can now be accepted for publication.

Experimental design

Additions made to the experimental design improved the overall comprehension of the methods e the design of the computational approach.

Validity of the findings

The findings are sound and grounded on excellent computational prediction capability.

Additional comments

None.